# Guidelines for the Evaluation of Cardiorespiratory Physiotherapy in Stroke Patients

**DOI:** 10.3390/healthcare8030222

**Published:** 2020-07-22

**Authors:** Sung-Hyoun Cho, Ki-Bok Choi

**Affiliations:** 1Department of Physical Therapy, Nambu University, 23 Cheomdanjungang-ro, Gwangsan-gu, Gwangju 62271, Korea; shcho@nambu.ac.kr; 2Team of Rehabilitation Treatment, Chosun University Hospital, 365 Pilmun-daero, Dong-gu, Gwangju 61453, Korea

**Keywords:** stroke, cardiorespiratory physiotherapy, modified Delphi survey, Likert scale, content validity ratio

## Abstract

Evaluation of stroke patients is prioritized over therapeutic interventions to restore cardiorespiratory capacity. This study aimed to develop a clinically applicable guideline to evaluate cardiorespiratory physiotherapy in stroke patients based on a literature review and a modified Delphi survey. The literature search included 13,498 articles in PubMed, EMBASE, CINAHL, and Cochrane Library electronic databases. We surveyed previous articles between January 2010 and June 2019. After the option elimination process, a total of 27 documents were selected and analyzed (draft: 18, modified Delphi survey: 9). The results of this research are roughly divided into two categories. First, 31 draft items were extracted, and a modified Delphi survey questionnaire was created from a literature review. Second, an expert was asked to make two modified Delphi surveys and to modify, delete, and supplement the entries in the statistical analysis at each level to finalize the steps to 20 items. The guidelines developed in this study reflect the selective use of cardiorespiratory physiotherapy evaluation methods in the clinical setting, based on the health status of individual patients. Moreover, the guidelines may help physiotherapists make informed decisions based on expert knowledge, thereby playing a crucial role in the patient-centered treatment planning process.

## 1. Introduction

The central nervous system regulates heart rate, cardiac contractility, blood pressure, and vasomotor tone, and diseases related to the regulation of autonomous blood flow occur after the onset of stroke [1]. Globally, stroke is the second leading cause of death and disability and is closely related to human mortality [2]. This necessitates an increased demand for rehabilitation programs to improve patient functional levels and quality of life.

A local brain injury in a stroke patient causes neurological damage, resulting in disorders such as a change in consciousness level, loss of motor and sensory function, and impaired cognitive and perceptual abilities. This results in limitations in functional movement and a decrease in physical activity [3]. Stroke requires a dynamic process of repair and remodeling to maintain neural circuits, which is shaped by the behavioral experience. Neuroplasticity depends on sufficient resolution of neurological dysfunction and restoration of blood flow, and the resolution of neural dysfunction may depend on neural plasticity [4]. The spontaneous and therapeutic-induced mechanisms of plasticity that promote functional activity along the site of the stroke injury provide a positive impact on recovery [5]. The rehabilitation strategy for stroke patients mainly focuses on posture and paralysis-related exercises [6]. However, focusing only on exercises that improve cardiac function is insufficient. As a result, physical activity is limited in most stroke patients, except during function-related therapy, which involves 1 h of therapy targeting the metabolic function of the heart and comprises 30 min of physiotherapy and 30 min of occupational therapy [7]. Thus, stroke patients have low cardiorespiratory endurance due to reduced physical activity [8]. Due to an inefficient energy metabolism, such as hemiplegic gait and spasticity, stroke patients exhibit increased muscle fatigue than the general population; incomplete cardiorespiratory control, abnormal oxygen transfer in the body, and decreased lung volume lead to an abnormal increase in the oxygen demand of the body and change in the cardiorespiratory function [9].

According to previous studies, the diaphragm and intercostal muscles, responsible for inhalation and exhalation in stroke patients, are mainly controlled by the unilateral cerebral cortex and subcortical spinal nerve pathways [10,11,12]. In particular, because the diaphragm is an important muscle responsible for approximately 10–70% of ventilation in the sitting and prone positions, a unilateral diaphragm paralysis can cause a significant decrease in respiratory function [13]. Based on this theory, previous studies have reported that for muscles on the paralyzed side, the gait speed correlates with the amount of oxygen intake, and muscle loss leads to low oxygen intake [14]. In addition, maximum oxygen intake, gait speed, and endurance in subacute stroke patients are less than 50% compared to those in healthy subjects [15].

For cardiac and respiratory rehabilitation, the American Heart Association (AHA) and the American Stroke Association (ASA) offer a multidisciplinary program that efficiently manages the exercise, diet, smoking, stress, and education of patients, and its psychological impact to reduce symptoms and help patients return to daily life [16,17]. Studies evaluating pulmonary function, cardiorespiratory endurance, and the 6-Minute Walk test (6MWT) have been actively conducted to evaluate the cardiorespiratory function of stroke patients [18,19]. However, there is still a lack of awareness of cardiorespiratory physiotherapy in stroke patients, and no studies have proposed a standardized guideline for evaluating cardiorespiratory physiotherapy conducted by researchers.

This study aimed to outline evaluation methods for cardiorespiratory physiotherapy in stroke patients, and a modified Delphi survey was used to achieve consensus between the literature review and an expert panel. The Delphi method, developed by the Research and Development (RAND) Corporation, is widely used to assist healthcare professionals to participate in developmental research by compensating for negative aspects such as time and location constraints [20].

Therefore, this study examined data from previous studies that evaluated cardiorespiratory physiotherapy in stroke patients and conducted a modified Delphi survey of experts’ opinions to develop clinically applicable guidelines and evaluation items for cardiorespiratory physiotherapy in stroke patients.

## 2. Materials and Methods

### 2.1. Search Strategy

To develop a guideline to evaluate cardiorespiratory physiotherapy in stroke patients, we systemically searched for articles published in accordance with the direction and purpose of this study between 1 January 2010 and 30 June 2019. 

We used the PubMed, EMBASE, CINAHL Plus with Full Text, and Cochrane Library databases. To increase the sensitivity and specificity of the literature search, operators (AND, OR) and truncated search (*) functions were used, and synonyms and alternate words of the keyword and Medical Subject Headings (MeSH), Emtree, and CINAHL subject headings were used (Table 1, Table 2, Table 3 and Table 4). In addition, in accordance with the Preferred Reporting Items for Systematic Reviews and Meta-Analyses (PRISMA), key questions in the system developed to evaluate cardiorespiratory physiotherapy in stroke patients (participants, intervention, comparisons, outcomes, study design [PICO-SD]) were selected [21].

### 2.2. Inclusion Criteria

In accordance with the PICO-SD, the subjects were adult stroke patients over 18 years of age, and the intervention measures included a single intervention (the single intervention group) or a combination of two or more interventions (the combination group). The results consisted of a single variable or two or more variables, including exercise interventions related to cardiac and respiratory physiotherapy. The study design included descriptive observational studies, case series; observational analytical, cross-sectional, case-control, and cohort studies; systematic reviews; meta-analyses; and randomized controlled trials (RCTs).

### 2.3. Exclusion Criteria

Stroke patients under the age of 18 years were excluded. In order to capture the latest research trends, the literature search to develop the draft guidelines was limited to studies published after 2010, but after the first and second modified Delphi surveys, we did not limit the year of publication in order to reflect the different opinions of experts. In addition, the language was limited to English.

### 2.4. Draft Development Process

A draft guideline was developed by listing the evaluation items to be considered in applying cardiorespiratory physiotherapy in stroke patients. The final selected research papers were independently reviewed by two researchers and one expert as the third figure.

### 2.5. Modified Delphi Process

#### 2.5.1. Ethical Considerations of Participants

Although this study has no human contact or human samples, it required the collection of personal information from the experts; therefore, the study was approved by the Institutional Review Board (IRB) of the Nambu University in Gwangju Metropolitan City, Korea (HBU-IRB-1041478-2017-HR-017) and was conducted in accordance with the ethical standards of the Declaration of Helsinki.

#### 2.5.2. Expert Selection

A panel of nine experts consisting of four full-time professors from the Department of Physical Therapy that oversaw academics and research in the field of cardiorespiratory physiotherapy and five clinical experts in charge of the clinical practice of cardiorespiratory physiotherapy was formed. We intended to fully reflect the opinions of experts in the development of clinically applicable guidelines.

The experts agreed to participate in the study and consisted of licensed physiotherapists with at least 10 years of clinical or research experience in cardiorespiratory physiotherapy. Most experts are active instructors in the Korean Academy of Cardiorespiratory Physiotherapy and are the core resources of cardiorespiratory physiotherapy in Korea.

#### 2.5.3. Modified Delphi Data Collection

The aim and method of the study were explained to the experts via phone beforehand. The panel consisted of cardiorespiratory physiotherapy experts that agreed to participate in the study (first trial: 9, second trial: 9). From 15 July, 2019 to 31 August, 2019, drafts using Likert scales and free text responses were sent to experts, and data consultation and content collection were conducted in person or via email.

#### 2.5.4. Confirmation of the Final Evaluation Items

The modified Delphi survey was repeated twice with the panel of experts, and after analyzing the survey results and reaching an agreement based on the statistical process, the final evaluation items were confirmed.

### 2.6. Data Analysis

The statistical analysis was conducted based on the data collected through the modified Delphi survey to the expert panel (first: 9 people, second: 9 people) using the SPSS version 20.0 statistical software (SPSS Inc., Chicago, IL, USA). In addition, descriptive statistical analysis was used to confirm the general characteristics of the expert panel and the verification of item understanding. In the modified Delphi survey, suitability of the items was measured using the Likert scale, with 1 point for “strongly disagree,” 2 points for “disagree”, 3 points for “neither agree nor disagree”, 4 points for “agree”, and 5 points for “strongly agree”, and the fitness values for each item were displayed.

Based on the displayed fitness values of the Likert scale, content validity and reliability were analyzed, and validity and suitability were calculated to revise and supplement the opinions of the expert panel.

#### 2.6.1. Content Validity and Reliability

To measure the accuracy and consistency of the measurement tool to judge the panel, the Content Validity Ratio (CVR) and Cronbach’s α coefficient were analyzed. In this study, based on 18 experts (9 in the first trial and 9 in the second trial), the minimum value of CVR was 0.78 [22].

#### 2.6.2. Validity and Stability

Convergence and consensus were analyzed to assess the validity of the surveys, which served as a criterion for judging the degree of convergence of the responses and the degree of consensus among the panel members. To determine convergence and consensus, the indicator of convergence had to approach 0 and that of consensus had to approach 1, for a given item to have a high level of validity [23,24]. In addition, stability was measured to judge the consensus among the panel of experts [25]; a coefficient of variation of 0.50 or less will lead to no additional survey, and 0.50 to 0.80 indicates that the survey result is relatively stable, whereas 0.80 or more necessitates an additional survey [26].

## 3. Results

### 3.1. Data Extraction Criteria

A literature search was carried out using the electronic databases, and the following number of articles were retrieved: PubMed (2789 articles), EMBASE (7399 articles), CINAHL Plus with Full text (1250 articles), and Cochrane library (2060 articles). A total of 13,498 articles were identified, and through a systematic screening of the collected articles following the procedure of the PRISMA statement and using Endnote software (X9.1, Clarivate Analytics. Philadelphia, PA, USA), the final dataset to be analyzed was selected (Figure 1). We identified 10,876 documents, excluding duplicate documents. Disagreements among reviewers were resolved by consensus; otherwise, a third reviewer was consulted. If not, it was resolved by a third external expert who is a physical therapist and statistical expert. Later, 18 articles were selected for draft development. The selected drafts were revised and analyzed by a panel of experts through the first and second modified Delphi surveys, after which 9 articles were added based on the liberal opinions of each panel member in the first and second surveys, leaving a final set of 27 articles. Table 5 and Table 6 present the results of the surveys conducted in this study.

### 3.2. General Characteristics of Selected Studies

Characteristics of the studies relating to the evaluation of cardiorespiratory physiotherapy in stroke patients were examined based on three main categories: the publication type, publication year, and study design (Table 7). Concerning the publication type, all 27 articles (100%) were published in academic journals. Concerning the publication year, 5 articles (18.5%) were published before 2010, 11 (40.7%) were published from 2011 to 2015, and 11 (40.7%) were published from 2016 to the present date (June 2019). With regard to the study design, 7 articles (25.9%) were experimental studies, 17 (63.0%) were observational studies, 2 (7.4%) were systematic reviews, and 1 (3.7%) was a systematic review with meta-analysis.

### 3.3. Draft Analysis

Based on the PRISMA statement, the 18 selected articles were analyzed for draft development by two investigators in this study and by a third external expert. Based on the title and abstract, studies related to the evaluation of cardiorespiratory physiotherapy were categorized according to the study design, and after compliance with the structure summary items based on the PICO-SD, duplicated studies were deleted; finally, a total of 31 draft items were considered (Table 8).

### 3.4. Modified Delphi Study Analysis 

#### 3.4.1. First Modified Delphi Study

In the first modified Delphi survey, the mean goodness of fit of the evaluation items was 4.05 ± 0.43 points based on the Likert scale responses. Regarding the content validity ratio, 16 items had a minimum score (less than 0.78) evaluated by the panel of 9 experts and were deleted following the analysis (i.e., mini-mental state examination (MMSE-K), visual analog scale (VAS), and Faces pain rate scale (FPRS), temperature test, light touch and pressure test, proprioception test, Albert test, range of motion (ROM), manual muscle test (MMT), Hand dynamometry, modified Ashworth scale, functional reach test (FRT), Uni-pedal stance test (UPST), Berg balance scale (BBS), dynamic gait index (DGI), modified Barthel index, and functional independence measure (FIM)). Internal consistency reliability for the evaluation items was set at α = 0.895. The mean convergence was 0.57, while the mean consensus and mean stability were 0.69 and 0.21, respectively (Table 9).

The results of the first modified Delphi survey showed that the accuracy of the statements for each item and the item validity exceeded the minimum scores; nevertheless, appropriate revision and modification were carried out to accommodate additional opinions from the panel and the items of low content validity (Table 10). Six items were newly formulated by the expert panel, including the body mass index (BMI) [45], spatiotemporal gait parameters [51], thoracic contour and chest expansion test [49], the trunk impairment scale (TIS) [52], cardiopulmonary exercise testing (CPET) [50], and the stroke impact scale (SIS) [48].

#### 3.4.2. Second Modified Delphi Study

The results of the first modified Delphi survey showed stability of 0.21 points and no additional questionnaire was required [26]. On the other hand, a degree of convergence close to 0 and a degree of consensus closer to 1 indicates that the opinions and consensus among the panels are valid [23,24]. According to the first modified Delphi survey, the mean convergence was 0.57 points and the mean consensus was 0.69 points, which were not properly reached and agreed among the panels. In addition, we needed to respond to open questions from expert panels; therefore, we conducted a second modified Delphi survey.

We conducted the second modified Delphi survey with the same panel of 9 experts, and the collection rate was 100%. The mean goodness of fit of the evaluation items was 4.40 ± 0.35 points based on the Likert scale responses. For the content validity ratio, two items with scores less than 0.78 (the minimum score of a panel of 9 experts) were deleted (Glasgow coma scale (GCS) and TIS). Concerning the validity, all other items exceeded the minimum score. The internal consistency reliability was α = 0.815. Likewise, the mean convergence improved from 0.57 to 0.36, the mean consensus improved from 0.69 to 0.84, and the mean stability improved from 0.21 to 0.17 (Table 11).

The accuracy of the statements for each item and the item validity all exceeded the minimum score; nevertheless, appropriate revision and modification were carried out to account for additional opinions from the panel and the items of low content validity (Table 12). The respiratory rate or minute ventilation [44] and the 10-Meter Walking Test (10MWT) [47] were added by expert panelists.

### 3.5. Deriving Final Evaluation Items

To evaluate cardiorespiratory physiotherapy in stroke patients, 31 evaluation items were derived from the literature review and were reorganized through the modified Delphi surveys by expert panelists. Twenty-two items were derived from the first and second survey to achieve consensus, the final 20 items were selected and divided among five main categories (Table 13).

## 4. Discussion

Appropriate clinical decision-making is a prerequisite to achieve adequate levels of evaluation as well as diagnosis, prognosis, and intervention based on the health status of individual patients [53]. However, clinical guidelines to evaluate cardiorespiratory physiotherapy in stroke patients are not yet available. The purpose of this study was to ensure enhanced quality and expertise for cardiorespiratory physiotherapy in stroke patients by providing selected evaluation items with verified evidence and promoting their use in the clinical setting.

As a result, we collected and analyzed 13,498 research papers, 18 articles were selected for the draft development, and after adding 9 articles via the modified Delphi surveys with expert panelists, a total of 27 articles were ultimately selected to determine the evaluation items. In the first modified Delphi survey, the expert panelists were requested to select from a wide range of evaluation items, including those extracted from the literature review on the comprehensive evaluation of stroke patients and those directly related to the cardiorespiratory domain. After statistically assessing the results of the first modified Delphi survey, individual items were assessed and those with a content validity less than 0.78 were deleted, while the remaining items were revised and modified based on stability, convergence, and consensus results and the open opinions provided by the experts to establish the second modified Delphi survey.

After the first modified Delphi survey, 16 items with less than the minimum score of 0.78 for content validity ratio, that is, the MMSE-K, VAS, FPRS, Temperature test, Light touch and pressure test, Proprioception test, Albert test, ROM, MMT, Hand dynamometry, modified Ashworth scale, FRT, UPST, BBS, DGI, modified Barthel index, and FIM, were deleted. The deleted items were found to be those required for primary motor function recovery without direct relation to cardiorespiratory fitness. In addition, the item internal consistency reliability of the first evaluation items was shown to be considerably high at α = 0.895. In determining convergence and consensus, the indicator of convergence should approach 0 and that of consensus should approach 1, for a given item to have a high level of validity [23,24]. The first modified Delphi survey result showed insufficient levels of convergence and consensus among the expert panelists with the mean convergence for the evaluation items at 0.57 and the mean consensus at 0.69. Regarding the stability, which indicates congruency among the experts in their responses, a coefficient of variation of 0.50 or less leads to no additional survey, 0.50 to 0.80 indicates that the survey result is relatively stable, whereas 0.80 or more necessitates an additional survey [26]. The mean stability was 0.21 in this study; therefore, no additional survey was required. However, due to insufficient convergence and consensus among the expert panelists and the need for expert responses for open questions, the second modified Delphi survey was conducted.

In this study, convergence of the open opinions of the expert panels was achieved using a method in which the investigator provided individual feedback to the issues raised as open opinions after the first modified Delphi survey. The open opinions provided by the expert panels included the need to clearly divide the categories of the main theme according to individual items, the need to extract the items through a multidisciplinary approach, and the need to categorize them according to the advancement of each phase of the stroke. The main feedback from the investigator was as follows: first, the evaluation items were divided into five categories: basic physical examination, level of consciousness, balance, cardiorespiratory, and activities of daily living; second, a guideline based on a multidisciplinary approach seemed an ideal method [54]; however, obtaining cooperation among health professionals in different fields was challenging, and the study anticipated that a considerable length of time and high cost were required. Thus, this study developed a guideline based on the findings of previous studies for cardiorespiratory evaluation items that were required for stroke patients; and third, based on the suggestion that the evaluation items should be provided according to the stroke phase (acute, subacute, and chronic), a practical method based on the health status of individual patients in the clinical setting seemed to be the ideal application for the guideline. The items added by the expert panelists included six evaluation items, namely, BMI [45], spatiotemporal gait parameters [51], the thoracic contour and chest expansion test [49], TIS [52], CPET [50], and SIS [48]. For the item of CPET in particular, the description in the draft limited the item to cycle graded exercise test (GXT), but based on a previous study reporting the use of CPET in treadmill and cycle methods, the item was added [16].

After the second modified Delphi survey, items GCS and TIS had less than the minimum score of 0.78 for content validity ratio and were deleted together with the cycle GXT that coincided with the CPET. In the basic physical examination category, respiratory rate or minute ventilation was added, while peak flow and ECG were removed from the basic physical examination category to the cardiorespiratory category. In the first modified Delphi survey, the pulmonary function test was evaluated as two separate items because of a large number of subcategories, but in the second survey, it was treated as a single unified item. Note that the pulmonary function test has been reported to be highly important as it provides basic data for precise exercise diagnosis by measuring accurate pulmonary function and assessing functional capacity, and the diagnosis and prognosis of the condition [55]. Comparing the convergence, consensus, and stability of the second modified Delphi survey results with those of the first modified Delphi survey, the mean convergence was shown to have improved from 0.57 to 0.36, the mean consensus from 0.69 to 0.84, and the mean stability from 0.21 to 0.17, which indicates that the convergence and consensus among the expert panelists increased further in the second modified Delphi survey with a higher level of congruency, suggesting that the experts reached a consensus. The added items were the respiratory rate or minute ventilation [44] and ten-meter walking test (10MWT) [47], implying that a consensus was drawn from the open opinions provided by the expert panelists.

In the second modified Delphi survey, the feedback to the liberal opinions of the expert panelists was as follows: First, the CPET methods include treadmill, leg cycle, and arm cycle exercises, and among these, the lower leg cycle CPET is considered the most efficient based on high levels of maximal heart rate (MHR) and respiratory gas exchange ratio (RER) as a risk of accidental falls. In addition, the treadmill-based CPET is a protocol of load-bearing exercises using the degree of slope, such that it is suggested to be more useful in achieving increased heart rate and anaerobic threshold compared to the 6MWT or incremental shuttle walk test (ISWT) speed [31]. Second, as the application of the ISWT and 6MWT is determined based on the health status of the patient, these items should be applied according to the patient’s health status in the clinical setting [46]. Third, for a treadmill walk exercise at 3 km/h, the energy expenditure is 2.43 metabolic equivalents (METs), and when the degree of slope is increased by 10% at the same speed to achieve higher exercise intensity, the energy expenditure is 5 METs. However, on a flat slope, even if the treadmill speed is increased two-fold to 6 km/h, the energy expenditure is only 3.85 METs, suggesting that it is more beneficial to stroke patients if the exercise intensity is increased by increasing the degree of the slope than by increasing the speed [56]. Fourth, it was determined that the Stroke Impact Scale (SIS) was a valid item from the Likert validity score of 4.11 ± 0.60. While the SIS seems to lack significant clinical evidence for its application, as no previous study has reported a definite association between stroke and cardiorespiratory parameters, it is possible that the items related to the movement correlate with mobility, activities of daily living (ADL)/instrumental activities of daily living (IADL), participation, and physical categories among the 9 categories, and above all, unlike the frequently used FIM and Barthel index, the SIS may be useful in determining the level of recovery for several symptoms of stroke [48]. Fifth, in the basic physical examination category, it was suggested that an additional respiratory test was required, such as the respiratory rate or minute ventilation [44]. Sixth, the 10MWT and 6MWT were suggested as additional items, as a previous study reported a significant correlation between the two tests with respect to physical activity and gait in stroke patients [57].

The present study has several limitations. First, when a given item is applied, the effect may vary according to the health status of individual stroke patients. This variation may be resolved through selective application of the proposed evaluation tool in a clinical setting. Second, the study did not take into consideration the impact of differences in clinical experience and academic background among the expert panelists. Third, the Physiotherapy Evidence Database (PEDro), a separate physical therapy database, was not added because the Cochrane Library database contains a wide range of papers and guidelines for randomized and systematic studies and clinical trials. It is highly advisable that this factor be analyzed and implemented in future studies.

In conclusion, the evaluation items suggested in this study were derived from converging evidence from expert opinions and should serve as an efficient tool in the clinical setting for the evaluation of cardiorespiratory physiotherapy necessity in stroke patients.

## 5. Conclusions

This study that actively reflects the opinions of expert panels with a modified Delphi survey that utilizes the advantages of non-face-to-face review and prior literature review to present items that can be applied to evaluate cardiorespiratory physiotherapy for stroke patients could be used directly in clinical practice. It is, therefore, important to develop an evidence-based guideline. The final 20 evaluation items can be widely used as a checklist for evaluating and diagnosing patients in clinical practice by selectively providing customized evaluation tools for cardiorespiratory physiotherapy with respect to the level of recovery of motor function in individual stroke patients. In addition, it is expected that it will have a positive impact on programs to plan therapeutic interventions by providing basic data for individual patient evaluations. Future experimental studies investigating the role of cardiorespiratory physiotherapy in stroke patients, in addition to systematic meta-analysis studies to allow a qualitative evaluation to secure its conclusions, are warranted to validate these guidelines.

## Figures and Tables

**Figure 1 healthcare-08-00222-f001:**
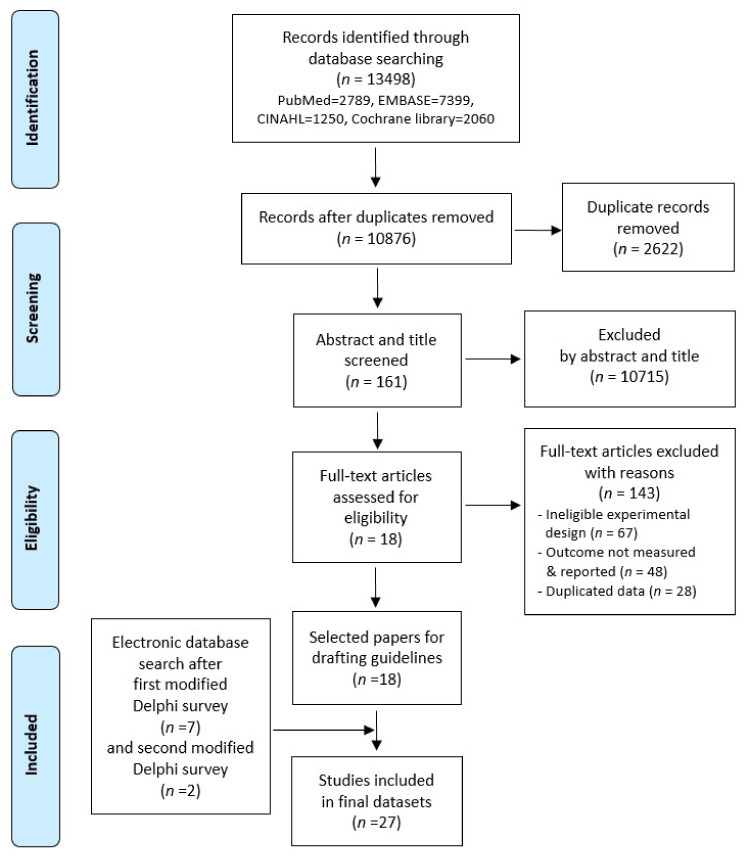
Flow diagram showing the literature inclusion and exclusion criteria in the Preferred Reporting Items for Systematic Reviews and Meta-Analyses (PRISMA) format.

**Table 1 healthcare-08-00222-t001:** Search strategy using Medical Subject Headings (MeSH) terms in PubMed.

Search Queries	Subject/MeSH Terms	
	User Group	Field of Health Research
Stroke AND Cerebral AND Exercise	Stroke Cerebrum Brain	Cardiac rehabilitation Respiratory therapy Patient outcome assessment Cardiorespiratory fitness Exercise Heart Rehabilitation
Stroke AND Cardiac rehabilitation AND Exercise		
Stroke AND Respiratory therapy AND Exercise		
Stroke AND Cardiorespiratory fitness AND Exercise		
Stroke AND Respirat * AND Exercise		
Stroke[abstract] AND Cerebral[abstract] AND Exercise[abstract] AND Rehabilitation[abstract] AND Respiratory[abstract] AND Cardiac[abstract]		
Stroke AND Cardiac rehabilitation AND Patient outcome assessments		
Stroke AND Cardiorespiratory fitness AND Patient outcome assessments		
Stroke AND Respirat * AND Patient outcome assessments		
Stroke[abstract] AND Cerebral[abstract] AND Respiratory[abstract] AND Cardiac[abstract] AND Patient outcome assessments[abstract]		
Stroke[title] AND Exercise[title] AND Respiratory[title]		
Stroke[title] AND Exercise[title] AND Cardiac[title]		
Hemiparetic AND Exercise		

Notes: Truncation search (*) is a comprehensive search method for searching terms.

**Table 2 healthcare-08-00222-t002:** Search strategy using Emtree terms in EMBASE.

ID	Search Queries
#1	‘Cerebrovascular disease’/exp AND ‘Stroke patient’/exp AND ‘Exercise’/exp
#2	‘Stroke patient’/exp OR ‘Cerebrovascular disease’/exp AND ‘Heart rehabilitation’/exp
#3	‘Stroke patient’/exp OR ‘Cerebrovascular disease’/exp AND ‘Respiratory care’/exp
#4	‘Stroke patient’/exp OR ‘Cerebrovascular disease’/exp AND ‘Cardiorespiratory fitness’/exp
#5	‘Stroke patient’/exp OR ‘Cerebrovascular disease’/exp AND ‘Breathing’/exp
#6	‘Stroke patient’/exp OR ‘Cerebrovascular disease’/exp AND ‘Breathing exercise’/exp
#7	‘Stroke patient’/exp OR ‘Cerebrovascular disease’/exp AND ‘Lung function test’/exp
#8	‘Stroke patient’/exp AND ‘Cerebrovascular disease’/exp AND ‘Outcome assessment’/exp
#9	‘Stroke patient’/exp OR ‘Cerebrovascular disease’/exp AND ‘Checklist’/exp
#10	‘Stroke patient’/exp OR ‘Cerebrovascular disease’/exp AND ‘Respiratory tract parameters’/exp AND ‘Breathing’/exp AND ‘Heart rehabilitation’/exp
#11	‘Stroke patient’/exp OR ‘Cerebrovascular disease’/exp AND ‘Exercise’/exp AND ‘Rehabilitation’/exp AND ‘Respiratory care’/exp

**Table 3 healthcare-08-00222-t003:** Search strategy in CINAHL.

ID	Search Queries
#1	Stroke AND Cerebral AND Exercise
#2	Stroke AND Cardiac rehabilitation AND Exercise
#3	Stroke AND Respiratory therapy AND Exercise
#4	Stroke AND Cardiorespiratory fitness AND Exercise
#5	Stroke AND Respirat * AND Exercise
#6	Stroke AND (MM “Breathing exercises”)
#7	Stroke AND (MH “Cardiorespiratory fitness”) OR (MH “Respiratory therapy equipment and supplies”) OR (MM “Respiratory therapy”) OR (MH “American association for respiratory care”)
#8	Stroke AND (MM “Outcome assessment”) OR (MM “Outcomes of education”) OR (MM “Patient assessment”)
#9	Stroke AND (MM “Outcome assessment”) OR (MM “Outcomes of education”) OR (MM “Patient assessment”) AND Pulmonary

Notes: Truncation search (*) is a comprehensive search method for searching terms.

**Table 4 healthcare-08-00222-t004:** Search strategy in the Cochrane Library.

ID	Search Queries
#1	Stroke AND Cerebral AND Exercise
#2	Stroke AND Cardiac rehabilitation AND Exercise
#3	Stroke AND Respiratory therapy AND Exercise
#4	Stroke AND Cardiorespiratory fitness AND Exercise
#5	Stroke AND Cardiac rehabilitation AND Patient outcome assessments
#6	Stroke AND Cardiorespiratory fitness AND Patient outcome assessments
#7	Stroke patient AND Cerebrovascular disease AND Outcome assessment
#8	Stroke patient AND Cerebrovascular disease AND Breathing exercise
#9	Stroke AND Cerebrovascular disease AND Exercise AND Rehabilitation AND Respiratory care
#10	Stroke patient AND Cerebrovascular disease AND Lung function test

**Table 5 healthcare-08-00222-t005:** Research characteristics selected for draft development.

#	Author (Country*)	Year	Journal	Design	Participants	Intervention	Comparison between Interventions	Outcome
1	Baert et al. [27] (Belgium)	2012	Arch Phys Med Rehabil	Descriptive, prospective longitudinal study	Sample of patients with stroke (*n* = 33)	CRF: VO_2peak_, OUES, workload peak, HR_peak_, RER_peak_, RPE scale	NA	CRF was reduced from 3 to 12 months poststroke and on average did not significantly change over time
2	Barclay et al. [28] (Canada)	2015	Cochrane Database Syst Rev	Systematic Reviews and meta-analysis	5 RCT studies (266 participants)	Community ambulation (practice, rehearsal, or exercise)	No treatment, usual treatment, other treatment, of placebo treatment	There is insufficient evidence to establish the effect of community ambulation interventions of the support a change in clinical practice
3	Boyne et al. [29] (USA)	2017	Top Stroke Rehabil	Cross-sectional observational study	Chronic stroke patients (*n* = 59)	Incline and speed GXT protocol, Inter-rater reliability testing for VO_2_-VT determination	NA	Motor dysfunction appears to artificially lower measured aerobic capacity. VO_2_-VT may provide more specific assessment of aerobic capacity post-stroke
4	Chang et al. [30] (Korea)	2012	Neurorehabil Neural Repair	Experimental study: Prospective single-blinded, RCT study	37 subacute stroke patients	Robot-assisted gait training (40 min) + conventional physical therapy (60 min)	Conventional physical therapy (100 min)	Patients can be trained to increase their VO_2_ and lower-extremity strength using a robotic device for stepping during inpatient rehabilitation
5	Dunn et al. [31] (Australia)	2019	Physiother Theory Pract	Cross-sectional observational study	Sample of patients with stroke (*n* = 23)	6MWT, ISWT, and GXT, during which VO_2peak_ and HR_peak_	NA	Correlations between VO_2peak_ and performance measures within each test were high
6	Dunn et al. [32] (Australia)	2017	Top Stroke Rehabil	Case-control observational study	17 SS participating	6MWT, ISWT, cycle GXT, walking speed, knee strength, body composition	17 healthy control participants	Peak VO_2_ was lower in the stroke group for the ISWT and progressive cycle test, as were all CRF test
7	Han and Im [33] (Korea)	2018	J Cardiopulm Rehabil Prev	Experimental study: RCT (Two-group pretest-posttest design)	Sample of patients with subacute stroke (*n* = 20)	ATE (30 min) for 6 weeks, five times per week	Land-based exercise for 6 weeks, five times per week	ATE group: significant improvements in 6MWT, VO_2peak_, HR_peak_, exercise tolerance test duration, and K-MBI
8	Harmsen et al. [34] (The Netherlands)	2017	Top Stroke Rehabil	Case series -descriptive study	Sample of patients with a-SAH (*n* = 27)	6MWT, CPET	NA	6MWT is an easy to administer submaximal exercise test in groups of patients with a-SAH
9	Marzolini et al. [35] (Canada)	2016	J Stroke Cerebrovasc Dis	Case series -descriptive study	Sample of post stroke patients (*n* = 60)	6MWT, CPET	NA	In combination with the CPET, the 6MWT will indicate when deficits preclude walking alone as the primary exercise modality for optimizing cardiovascular fitness
10	Menezes et al. [36] (Brazil)	2018	Respir Care	Systematic Reviews	17 RCT studies (616 stroke patients)	Interventions to improve respiratory function	Control group	Meta-analyses found no significant results for the effects of breathing exercise on lung function
11	Moura et al. [37] (Brazil)	2015	Neurol Int	Case series-descriptive study	10 hemiparetic patients diagnosed with CVA (*n* = 10)	The correlation between the 6MWD and the biomechanical profile of hemiparetic patients	NA	The negative correlation between the percentage of loss of 6MWD and the limitation in the ankle dorsiflexion movement
12	Ovando et al. [38] (Brazil)	2011	Arq Bras Cardiol	Cross-sectional observational study	8 individuals with chronic hemiparesis	Evaluation of cardiopulmonary test (CPT)	NA	The CPT proved to be useful for prescribing physical activity in these individuals
13	Paz [39] (Spain)	2016	Arch Phys Med Rehabil	Cross-sectional observational study	30 SS with 30 healthy people matched by age and sex	Respiratory muscle strength (MEP and MIP), 6MWT, MI, SIS-16	NA	MIP and MEP are significantly decreased in SS. Inspiratory and expiratory muscle weakness was observed as a clinically relevant finding
14	Salbach et al. [40] (Canada)	2014	Neurorehabil Neural Repair	Case series -descriptive study	Consecutive sample of post-stroke patients (*n* = 16)	Comparing cardiorespiratory responses to CET and 6MWT	NA	Exercise intensity achieved during the 6MWT appeared sufficiently high for aerobic training, assuming CET VO_2peak_ accurately reflects aerobic capacity
15	Söderholm et al. [41] (Sweden)	2012	Stroke	Prospective cohort study	20,534 men and 7237 women	Lung function as a risk factor (incidence of SAH, smoking, and hypertension) for SAH	NA	Baseline lung function, expressed as low FEV_1_ or FEV_1_/FVC, is a risk factor for SAH, independently of smoking
16	Smith et al. [42] (UK)	2012	Int J Stroke	Systematic Reviews	41 studies (RCT: 16, UCT: 7, NRCT: 1, Obs: 14, Cohort Obs: 3)	Maximal oxygen uptake in SS	Maximal oxygen uptake with published data from age- and sex-matched controls	CRF levels are significantly lower in patients with stroke compared with healthy people of same age and gender
17	Tang et al. [43] (Canada)	2010	BMC Neurol	Cohort observational study	Sample of patients with stroke who completed the study (*n* = 38)	Feasibility (number of participants who completed the study, occurrence of adverse events and frequency, duration, and intensity of exercise), Effectiveness (aerobic capacity, 6MWT distance, and risk factors)	Aerobic capacity (VO_2peak_, ventilatory threshold), 6MWT distance, and risk factors (after 6 months)	CR is feasible after stroke and may be adapted to accommodate for those with a range of post-stroke disability
18	Woodward et al. [18] (USA)	2019	Phys Ther	Cross-sectional observational study	Sample of patients with chronic (>6 months) hemiparesis (*n* = 53)	Cardiorespiratory responses during 6MWTs and GXTs	NA	Cardiac responses were higher than anticipated during 6MWTs and often exceeded recommended HR thresholds

Country* = refers to the location of the corresponding author. Abbreviations: 6MWD = Six-minute walk distance; 6MWT = Six-minute walk test; a-SAH = aneurysmal Subarachnoid hemorrhage; AIT = Aerobic interval training; ATE = Aquatic treadmill exercise; CET = Cycle ergometer test; CPET = Cardiopulmonary exercise testing; CPT = Cardiopulmonary testing; CRF = Cardiorespiratory fitness; FEV_1_ = Forced expiratory volume in 1 s; FVC = Forced vital capacity; GXT = Graded exercise test; HIIT = High-intensity interval training; HR = Heart rate; ISWT, Incremental shuttle walk test; MEP = Maximal expiratory pressure; MI = Motricity index; MIP = Maximal inspiratory pressure; NA = Not applicable; NRCT = Nonrandomized controlled trial; Obs = Observational study; OUES = Oxygen uptake efficiency slope; RCT = Randomized controlled trial; RER = Respiratory gas exchange ratio; RPE = Ratings of perceived exertion; SAH = Subarachnoid hemorrhage; SIS-16 = Impact of stroke version 16.0; SS = Stroke survivors; TAEX = Aerobic treadmill exercise; UCT = Uncontrolled trial; VO_2_ = Oxygen consumption; VO_2peak_ = Peak oxygen uptake; VO_2_-VT = VO_2_ at the ventilatory threshold.

**Table 6 healthcare-08-00222-t006:** Research characteristics selected after the first and second modified Delphi investigations surveys.

#	Author (Country*)	Year	Journal	Design	Participants	Intervention	Comparison between Interventions	Outcome
1	Betensley et al. [44] (USA)	2008	Respir Care	Randomized prospective trial	14 patients with intubated mechanically ventilated patients	Patient comfort: 100 cm visual analog scale (PRVC ventilation, PSV or VC-CMV)	NA	On average the patients felt more comfortable during PSV than during VC-CMV or PRVC, so PSV may be the preferred mode for awake intubated patients
2	Chen et al. [45] (Taiwan)	2018	Kaohsiung J Med Sci	Experimental study: Single-group, pre-post test, intervention study	37 acute stroke patients (Stroke onset less than two weeks)	Early stroke rehabilitation (individualized treatment program) for three weeks	NA	Improvement of functional outcome (FMS, FIM) and Increase of cardiovascular fitness (peak workload, oxygen uptake, heart rate, oxygen pulse)
3	Clague-Baker et al. [46] (UK)	2019	Physiotherapy	Cross-sectional observational study	40 stroke patients within six months of stroke	Validity between VO_2Peak_, ISWT and 6MWT, Test re-test reliability of 6MWT and ISWT	NA	The ISWT and 6MWT have a significant, modest correlation with the ICT for stroke patients in the subacute recovery phase
4	Dalgas et al. [47] (Denmark)	2012	Arch Phys Med Rehabil	Cross-sectional observational study	Patients with multiple sclerosis (*n* = 38), patients with stroke (*n* = 48), and healthy subjects (*n* = 46)	Correlations between walking tests, relationship between walking speed from long and short walking tests	NA	Walking speeds of a short walking test and a long walking test are strongly correlated in Multiple sclerosis and stroke, whereas correlations in healthy subjects are weaker
5	Duncan et al. [48] (USA)	1999	Stroke	Prospective cohort study	Subjects with mild and moderate strokes completed the SIS at 1 month (*n* = 91), at 3 months (*n* = 80), and at 6 months after stroke (*n* = 69)	Internal consistency and test-retest reliability, the validity	NA	This new, stroke-specific outcome measure is reliable, valid, and sensitive to change
6	Kim et al. [49] (Korea)	2011	J Phys Ther Sci	Experimental study: RCT (Two-group pretest-posttest control group design)	Chronic stroke patients (*n* = 27)	Feedback respiratory training (30 min) and Conventional physical therapy (30 min) for 4 weeks, three times per week	Conventional physical therapy (30 min) for 4 weeks, three times per week	Feedback respiratory training is effective for the improvement of chest expansion and pulmonary function
7	Kim et al. [50] (Korea)	2014	Med Sci Monit	Experimental study: RCT (Two-group pretest-posttest control group design)	Stroke patients (*n* = 20)	The same exercise regimen as the control group (30 min) and respiratory muscle training regimen using a respiratory exercise device (20 min), 3 times per week for 4 weeks	Control group: Basic exercise treatments for 30 min, followed by an automated full-body workout, 3 times per week for 4 weeks	Exercise of the respiratory muscles using an individualized respiratory device had a positive effect on pulmonary function and exercise capacity
8	Tang et al. [51] (Canada)	2009	Neurorehabil Neural Repair	Prospective matched control design	45 subacute stroke patients (less than 3 months after stroke)	30 min of aerobic cycle ergometry + conventional inpatient rehabilitation 3 days/weeks until discharge	Conventional rehabilitation	Improvement of aerobic capacity, Gait assessment, 6MWT, SIS
9	Verheyden et al. [52] (Belgium)	2004	Clin Rehabil	Cross-sectional observational study	Chronic stroke patients (*n* = 28)	Kappa and ICC of Test/retest agreement, Inter-observer agreement	NA	Guidelines for treatment and level of quality of trunk activity can be derived from the assessment

Country* = means the location of corresponding author. Abbreviations: 6MWT = Six-minute walk test; FIM = Functional independence measure; FITT = Frequency and intensity and time and type; FMS = Fugl-Meyer scale; ICC = Intraclass correlation; ICT = Incremental cycle test; ISWT = Incremental shuttle walk test; NA = Not applicable; PRVC = Pressure-regulated volume-control; PSV = Pressure-support ventilation; RCT = Randomized controlled trial; SIS = Stroke impact scale; VC-CMV = Volume controlled-continuous mandatory ventilation; VO_2peak_ = Peak oxygen uptake.

**Table 7 healthcare-08-00222-t007:** Characteristics of the analyzed studies (*n* = 27).

Main Category	Sub-Category	No (%) of Reports >
Publication year	~2010	5 (18.6%)
	2011~2015	11 (40.7%)
	2016~present	11 (40.7%)
Publication type	Journal	27 (100%)
Study design	Experimental study	7 (25.9%)
	Observational study	17 (63.0%)
	Systematic reviews	2 (7.4%)
	Systematic reviews and Meta-analysis	1 (3.7%)

**Table 8 healthcare-08-00222-t008:** Draft item.

No	Draft Item
1	Vital sign
2	Saturation
3	Peak flow
4	ECG
5	LOC
6	GCS
7	MMSE-K
8	VAS and FPRS
9	Temperature test
10	Light touch and pressure test
11	Proprioception test
12	Albert test
13	ROM test
14	MMT
15	MIP and MEP
16	Hand dynamometry
17	MAS
18	FRT
19	UPST
20	BBS
21	DGI
22	TUG
23	RPE
24	modified Borg dyspnea scale
25	6MWT
26	ISWT
27	Pulmonary function test
28	Cycle GXT
29	SGRQ
30	MBI
31	FIM

Abbreviations: 6MWT = Six-minute walk test; BBS = Berg balance scale; DGI = Dynamic gait index; ECG = Electrocardiographic; FIM = Functional independence measure; FPRS = Faces pain rate scale; FRT = Functional reach test; GCS = Glasgow coma scale; GXT = Graded exercise test; ISWT = Incremental shuttle walk test; LOC = Level of consciousness; MAS = modified Ashworth scale; MBI = modified Barthel index; MEP = Maximal expiration pressure; MIP = Maximal inspiration pressure; MMT = Manual muscle test; MMSE-K = Mini-mental state examination; ROM = Range of motion; RPE = Rating of perceived exertion; TUG = Timed up and go test; UPST = Uni-pedal stance test; SGRQ = Saint George’s respiratory questionnaire; VAS = Visual analog scale.

**Table 9 healthcare-08-00222-t009:** Results of items after the first modified Delphi investigation.

No	Evaluation Item	Mean	SD	CVR	Q1	Median	Q3	Stability	Convergence	Consensus
1	Vital sign	4.89	0.33	1.00	5.00	5.00	5.00	0.07	0.00	1.00
2	Saturation	5.00	0.00	1.00	5.00	5.00	5.00	0.00	0.00	1.00
3	Peak flow	4.56	0.73	0.78	4.00	5.00	5.00	0.16	0.50	0.80
4	ECG	5.00	0.00	1.00	5.00	5.00	5.00	0.00	0.00	1.00
5	LOC	4.33	0.71	0.78	4.00	4.00	5.00	0.16	0.50	0.75
6	GCS	4.22	0.67	0.78	4.00	4.00	5.00	0.16	0.50	0.75
7	MMSE-K	3.78	0.67	0.33	3.00	4.00	4.00	0.18	0.50	0.75
8	VAS and FPRS	3.78	0.97	−0.11	3.00	3.00	5.00	0.26	1.00	0.33
9	Temperature sensory test	2.78	0.97	−0.56	2.00	3.00	3.50	0.35	0.75	0.50
10	Light touch and pressure test	2.67	1.00	−0.56	2.00	3.00	3.50	0.38	0.75	0.50
11	Proprioception sensory test	3.56	1.01	0.33	2.50	4.00	4.00	0.29	0.75	0.63
12	Albert test	3.22	0.97	−0.33	2.50	3.00	4.00	0.30	0.75	0.50
13	ROM test	3.67	1.12	0.33	2.50	4.00	4.50	0.30	1.00	0.50
14	MMT	3.44	1.13	−0.11	2.50	3.00	4.50	0.33	1.00	0.33
15	MIP and MEP	4.78	0.44	1.00	4.50	5.00	5.00	0.09	0.25	0.90
16	Hand dynamometry	4.22	1.09	0.56	3.50	5.00	5.00	0.26	0.75	0.70
17	MAS	3.33	1.00	−0.11	2.50	3.00	4.00	0.30	0.75	0.50
18	FRT	3.22	0.97	0.11	2.00	4.00	4.00	0.30	1.00	0.50
19	UPST	3.33	1.22	0.11	2.50	4.00	4.00	0.37	0.75	0.50
20	BBS	3.89	1.05	0.33	3.00	4.00	5.00	0.27	1.00	0.63
21	DGI	3.33	1.41	0.11	2.00	4.00	4.50	0.42	1.25	0.38
22	TUG	4.22	1.30	0.78	4.00	5.00	5.00	0.31	0.50	0.80
23	RPE	4.78	0.44	1.00	4.50	5.00	5.00	0.09	0.25	0.90
24	Modified Borg dyspnea scale	4.89	0.33	1.00	5.00	5.00	5.00	0.07	0.00	1.00
25	6MWT	4.78	0.67	0.78	5.00	5.00	5.00	0.14	0.00	1.00
26	ISWT	4.33	0.71	0.78	4.00	4.00	5.00	0.16	0.50	0.75
27	Pulmonary function test	5.00	0.00	1.00	5.00	5.00	5.00	0.00	0.00	1.00
28	Cycle GXT	4.56	0.73	0.78	4.00	5.00	5.00	0.16	0.50	0.80
29	SGRQ	4.56	0.53	1.00	4.00	5.00	5.00	0.12	0.50	0.80
31	FIM	3.56	1.01	−0.11	3.00	3.00	4.50	0.29	0.75	0.50
Average		4.05	0.43	0.45	3.50	4.19	4.65	0.21	0.57	0.69

Notes: The shadings are deleted items after first modified Delphi investigation. Abbreviations: 6MWT = Six-minute walk test; BBS = Berg balance scale; CVR = Content validity ratio; DGI = Dynamic gait index; ECG = Electrocardiographic; FIM = Functional independence measure; FPRS = Faces pain rate scale; FRT = Functional reach test; GCS = Glasgow coma scale; GXT = Graded exercise test; ISWT = Incremental shuttle walk test; LOC = Level of consciousness; MAS = modified Ashworth scale; MBI = modified Barthel index; MEP = Maximal expiration pressure; MIP = Maximal inspiration pressure; MMT = Manual muscle test; MMSE-K = Mini-mental state examination; Q1 = Lower quartile; Q3 = Upper quartile; ROM = Range of motion; RPE = Rating of perceived exertion; SD = Standard deviation; TUG = Timed up and go test; UPST = Uni-pedal stance test; SGRQ = Saint George’s respiratory questionnaire; VAS = Visual analog scale.

**Table 10 healthcare-08-00222-t010:** Open opinions and researcher feedback after the first modified Delphi survey.

Open Opinion Content	Feedback Content
To address the need for clearly dividing the categories of the main theme according to individual items.	-The evaluation items were divided into five categories: basic physical examination, level of consciousness, balance, cardiorespiratory, and activities of daily living.
To extract the evaluation items through a multidisciplinary approach.	-Cooperation among health professionals of different fields was challenging. -A considerable length of time and high cost were anticipated. -Thus, a guideline has been developed in this study based on the findings of previous studies, for cardiorespiratory evaluation and intervention items that are required for stroke patients.
To extract the evaluation items according to the advancement of each phase of stroke.	-The evaluation and intervention items showed differences according to the health status of individual patients, which are problematic. -A practical method in the clinical setting seemed to be selective application based on the guideline developed in this study, according to each patient’s health status.
To include the description of factors to consider for the discontinuation of exercise by adding an evaluation parameter related to chest pain and leg pain.	-For stroke, the rationale for discontinuation of exercise did not derive from a specific pain as in patients with angina or peripheral vascular disease. -In general, discontinuation of exercise depended on the result of the basic indices such as SPO_2_, HR, and BP, or patient’s complaints on the symptoms of headache, vertigo, nausea, and vomit.
To include the description on additional items for the CPET.	-In this study, due to a possibility of falls in stroke patients, GXT description by the cycle had been included based on previous studies; nevertheless, based on a previous study reporting the use of CPET in stroke patients, the item was added.
To add BMI in the basic physical examination category, as it is always included as a test item in cardiorespiratory tests and interventions, and as it is one of the important factors.	-The BMI was added to the category of basic physical examination in evaluation items.
To add spatiotemporal gait parameters used as an indicator of balance in the category of exercise for stroke patients.	-The spatiotemporal gait parameters were added to the category of balance in evaluation items.
To address the need for FVC test, as the stroke-related breathing difficulty is likely to be caused by restrictive lung disease due to reduced lung compliance. To add the Thoracic contour and Chest expansion test, for an asymmetric thoracic cage may lead to local lung collapse. In addition, only after chest expansion test, the lobe and area of the lung with restricted expansion can be detected, for which a suitable chest expansion method or chest mobilization, or segmental breathing for local lung inflation, can be determined.	-FVC test was added as a pulmonary function test, as well as the items; Thoracic contour and Chest expansion test. -The items; chest mobilization and segmental breathing, were added.
The items; MIP and MEP, should be moved to the cardiorespiratory category as they test the respiratory capacity, and the static lung capacity tests (TV, IRV, ERV, VC, IC) should be added.	-For the pulmonary function test, the items, SVC and maximal-effort expiratory capacity, were revised and subsequently separated. -The items; MIP and MEP, were moved to the cardiorespiratory category.
Cervical and thoracic mobilization.	-Cervical and thoracic mobilization is anticipated to be more helpful during the intervention, rather than limiting to chest mobilization.

Abbreviations: BMI = Body mass index; CPET = Cardiopulmonary exercise test; ERV = Expiratory reserve volume; FVC = Forced vital capacity; GXT = Graded exercise test; IC = Inspiratory capacity; IRV = Inspiratory reserve volume; MEP = Maximal expiration pressure; MIP = Maximal inspiration pressure; SVC = Slow vital capacity; TV = Tidal volume; VC = Vital capacity.

**Table 11 healthcare-08-00222-t011:** Results of items after the second modified Delphi investigation.

Domain	No	Evaluation Item	Mean	SD	CVR	Q1	Median	Q3	Stability	Convergence	Consensus
Basic physical examination	1	BMI	4.67	0.71	0.78	4.50	5.00	5.00	0.15	0.25	0.90
	2	Vital sign	4.89	0.33	1.00	5.00	5.00	5.00	0.07	0.00	1.00
	3	Saturation	4.89	0.33	1.00	5.00	5.00	5.00	0.07	0.00	1.00
	4	Peak flow	4.56	0.53	1.00	4.00	5.00	5.00	0.12	0.50	0.80
	5	ECG	4.56	1.33	0.78	5.00	5.00	5.00	0.29	0.00	1.00
Level of consciousness	6	LOC	4.33	0.50	1.00	4.00	4.00	5.00	0.12	0.50	1.00
	7	GCS	3.89	0.93	0.56	3.50	4.00	4.50	0.24	0.50	0.75
Balance	8	Spatiotemporal gait parameters	4.22	0.67	0.78	4.00	4.00	5.00	0.16	0.50	0.75
	9	TUG	4.56	0.53	1.00	4.00	5.00	5.00	0.12	0.50	0.80
Cardiorespiratory	10	Thoracic contour and Chest expansion test	4.22	0.97	0.78	4.00	4.00	5.00	0.23	0.50	0.75
	11	TIS	3.56	1.01	0.33	2.50	4.00	4.00	0.29	0.75	0.63
	12	MIP and MEP	4.56	0.53	1.00	4.00	5.00	5.00	0.12	0.50	0.80
	13	RPE	4.67	0.50	1.00	4.00	5.00	5.00	0.11	0.50	0.80
	14	mMRC	4.67	0.50	1.00	4.00	5.00	5.00	0.11	0.50	0.80
	15	6MWT	4.56	0.53	1.00	4.00	5.00	5.00	0.12	0.50	0.80
	16	ISWT	3.78	1.09	0.78	4.00	4.00	4.00	0.29	0.00	1.00
	17	CPET	4.00	1.22	0.78	4.00	4.00	5.00	0.31	0.50	0.75
	18	Cycle GXT	4.00	1.22	0.78	4.00	4.00	5.00	0.31	0.50	0.75
	19	Pulmonary function test 1. SVC	4.78	0.44	1.00	4.50	5.00	5.00	0.09	0.25	0.90
	20	Pulmonary function test 2. Maximal-effort expiratory capacity	4.89	0.33	1.00	5.00	5.00	5.00	0.07	00.00	1.00
Activities of Daily Living	21	SGRQ	4.44	0.73	0.78	4.00	5.00	5.00	0.16	0.50	0.80
	22	SIS	4.11	0.60	0.78	4.00	4.00	4.50	0.15	0.25	0.88
Average			4.40	0.35	0.86	4.14	4.59	4.86	0.17	0.36	0.84

Notes: The shadings are deleted items after second modified Delphi investigation. Abbreviations. Abbreviations: 6MWT = Six-minute walk test; BMI = Body mass index; CPET = Cardiopulmonary exercise test; CVR = Content validity ratio; ECG = Electrocardiographic; GCS = Glasgow coma scale; GXT = Graded exercise test; ISWT = Incremental shuttle walk test; LOC = Level of consciousness; MEP = Maximal expiration pressure; MIP = Maximal inspiration pressure; mMRC = modified Medical research council; Q1 = Lower quartile; Q3 = Upper quartile; RPE = Rating of perceived exertion; SD = Standard deviation; SGRQ = Saint George’s respiratory questionnaire; SIS = Stroke impact scale; SVC = Slow vital capacity; TIS = Trunk impairment scale; TUG = Timed up and go test.

**Table 12 healthcare-08-00222-t012:** Second survey after the second modified Delphi investigation, open opinions, and researcher feedback.

Open Opinion Content	Feedback Content
To combine the items; CPET and cycle GXT	-The CPET includes the following methods: treadmill, leg cycle, and arm cycle, and among these, the lower leg cycle CPET shows high levels of MHR and RER, which is likely to be useful in evaluating patients with a risk of falls.
	-The treadmill-based CPET is a protocol of load-bearing exercise using the degree of slope, so that it is likely to prove more useful in achieving increased heart rate and anaerobic threshold in comparison to 6MWT or ISWT using speed.
To suggest a method of application regarding ISWT and 6MWT	-The use of ISWT and 6MWT is determined according to the patient’s health status, so that in the clinical setting, it is likely to be more efficient to apply based on the individual patient.
To consider inclusion of SIS items	-Among the 9 categories, a correlation of the mobility, ADL/IADL, participation, and physical categories, for items related to movement, and above all, unlike the frequently used FIM and Barthel index, SIS items were likely to prove useful in determining the level of recovery for several symptoms of stroke in clinical setting.
To add the item; Respiratory rate or Minute ventilation, in the category of basic physical examination.	-The items were added because the category of basic physical examination required a respiratory parameter such as respiratory rate or minute ventilation.
To add the item; 10MWT	-10MWT and the 6MWT were added, as a previous study reported a significant correlation between the two tests with respect to the physical activity and gait in stroke patients.
To apply Thoracic contour and Chest expansion test	-The two items are rarely applied in clinical setting; thus, further investigations are required.
To add an item related to quality of life with respect to the health of the heart	-No study was found for the suggested parameter that can be applied to stroke patients.

Abbreviations: 6MWT = Six-minute walk test; 10MWT = Ten-meter walking test; ADL = Activities of daily living; CPET = Cardiopulmonary exercise test; FIM = Functional independence measure; GXT = Graded exercise test; IADL = Instrumental activities of daily living; ISWT = Incremental shuttle walk test; MHR = Maximal heart rate; RER = Respiratory exchange ratio; SIS = Stroke impact scale.

**Table 13 healthcare-08-00222-t013:** Cardiorespiratory physiotherapy final evaluation items for stroke patients.

Domain	No	Final Item
Basic physical examination	1	BMI
	2	Vital sign
	3	Saturation
	4	Respiratory rate or Minute ventilation
Level of consciousness	5	LOC
Balance	6	Spatiotemporal gait parameters
	7	10MWT
	8	TUG
Cardiorespiratory	9	Peak flow
	10	ECG
	11	Thoracic contour and Chest expansion test
	12	MIP and MEP
	13	RPE
	14	mMRC
	15	6MWT
	16	ISWT
	17	CPET
	18	Pulmonary function test 1. SVC (1) TV (2) ERV (3) IRV (4) IC (5) VC 2. Maximal-effort expiratory capacity (1) FVC (2) FEV_1_ (3) FEV_1_/FVC (4) FEF 25~75%
Activities of Daily Living	19	SGRQ
	20	SIS

Abbreviations: 6MWT = Six-minute walk test; 10MWT = Ten-meter walking test; BMI = Body mass index; CPET = Cardiopulmonary exercise test; ECG = Electrocardiographic; ERV = Expiratory reserve volume; FEF = Forced expiratory flow; FEV_1_ = Forced expiratory volume in 1 s, FVC = Forced vital capacity; IC = Inspiratory capacity; IRV = Inspiratory reserve volume; ISWT = Incremental shuttle walk test; LOC = Level of consciousness; MEP = Maximal expiration pressure; MIP = Maximal inspiration pressure; mMRC = modified Medical research council; RPE = Rating of perceived exertion; SGRQ = Saint George’s respiratory questionnaire; SIS = Stroke impact scale; SVC = Slow vital capacity; TUG = Timed up and go test; TV = Tidal volume; VC = Vital capacity.

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
