# Peer review of "Guidelines for the Evaluation of Cardiorespiratory Physiotherapy in Stroke Patients"

_healthcare, 2020, doi:10.3390/healthcare8030222_

Round 1

Reviewer 1 Report

Thank you very much for allowing me to review the manuscript. The topic is very interesting, and I think it will be very useful for clinical practice.
I will now make some recommendations for improving the quality of the manuscript.
- Good introduction, explaining in detail the problem to be addressed.
- Figure 1 shows two references with the same number of results. please clarify this situation
- It would be good to improve the exposition of the statistical part of the study.
- Clarify the role of the third reviewer. It has been mentioned several times in the text, but it has not been sufficiently clear.
- It would be good to decrease the number of tables so that it is not exceeded and can complicate the reading and understanding of the study.
- Better clarify why it does double Delphi analysis
- I think it would be more appropriate to review the results section, so that the reader can understand in a more synthesized way the most important results obtained in the search.
- In the conclusion section, do not repeat information previously given, but be more specific.
- Check references number: 4, 5, 8, 10, 19 and 44 because they are not updated.
- Reference number 21 appropriately (according to publication regulations)

Author Response

We attached the response file.

Reviewer 2 Report

According to the attached report.

Author Response

We attached the file.
